# Neurotoxic Effect of Doxorubicin Treatment on Cardiac Sympathetic Neurons

**DOI:** 10.3390/ijms231911098

**Published:** 2022-09-21

**Authors:** Nicola Moro, Lolita Dokshokova, Induja Perumal Vanaja, Valentina Prando, Sophie Julie A Cnudde, Anna Di Bona, Riccardo Bariani, Leonardo Schirone, Barbara Bauce, Annalisa Angelini, Sebastiano Sciarretta, Alessandra Ghigo, Marco Mongillo, Tania Zaglia

**Affiliations:** 1Department of Biomedical Sciences, University of Padova, via Ugo Bassi 58/B, 35131 Padova, Italy; 2Department of Cardiac, Thoracic, Vascular Sciences and Public Health, University of Padova, via Giustiniani 2, 35128 Padova, Italy; 3Molecular Biotechnology Center, Department of Molecular Biotechnology and Health Sciences, University of Torino, 10126 Torino, Italy; 4Department of Medical and Surgical Sciences and Biotechnologies, Sapienza, University of Rome, 04100 Latina, Italy

**Keywords:** Doxorubicin, cardiotoxicity, sympathetic neurons, cardiac innervation, nerve growth factor

## Abstract

Doxorubicin (DOXO) remains amongst the most commonly used anti-cancer agents for the treatment of solid tumors, lymphomas, and leukemias. However, its clinical use is hampered by cardiotoxicity, characterized by heart failure and arrhythmias, which may require chemotherapy interruption, with devastating consequences on patient survival and quality of life. Although the adverse cardiac effects of DOXO are consolidated, the underlying mechanisms are still incompletely understood. It was previously shown that DOXO leads to proteotoxic cardiomyocyte (CM) death and myocardial fibrosis, both mechanisms leading to mechanical and electrical dysfunction. While several works focused on CMs as the culprits of DOXO-induced arrhythmias and heart failure, recent studies suggest that DOXO may also affect cardiac sympathetic neurons (cSNs), which would thus represent additional cells targeted in DOXO-cardiotoxicity. Confocal immunofluorescence and morphometric analyses revealed alterations in SN innervation density and topology in hearts from DOXO-treated mice, which was consistent with the reduced cardiotropic effect of adrenergic neurons in vivo. Ex vivo analyses suggested that DOXO-induced denervation may be linked to reduced neurotrophic input, which we have shown to rely on nerve growth factor, released from innervated CMs. Notably, similar alterations were observed in explanted hearts from DOXO-treated patients. Our data demonstrate that chemotherapy cardiotoxicity includes alterations in cardiac innervation, unveiling a previously unrecognized effect of DOXO on cardiac autonomic regulation, which is involved in both cardiac physiology and pathology, including heart failure and arrhythmias.

## 1. Introduction

Cancer accounts for more than 8 million deaths per year. Doxorubicin (DOXO) is one of the most common anti-cancer agents, used for the treatment of several solid tumors, lymphomas, and leukemias. Unfortunately, a major complication of DOXO regimens is represented by cardiotoxicity [1,2,3,4,5], characterized by heart failure (HF) and increased arrhythmic vulnerability, which may require chemotherapy interruption, with devastating consequences on tumor progression and increased mortality [4,6]. Remarkably, HF is the first cause of mortality in cancer survivors [7,8,9,10,11]. Despite intense research efforts, the mechanisms underlying DOXO-induced cardiotoxicity are still largely nebulous and strategies to treat the adverse consequences of the drug are lacking. Thus, cancer patients are in urgent need of effective responses to prevent a life-saving therapy from marking the onset of a new deadly disease. In addition, although the link between DOXO and malignant arrhythmias is consolidated [12,13], the underlying mechanisms are still obscure. 

We and others recently demonstrated that DOXO leads to heart dysfunction [14,15,16,17]. In addition, a set of studies demonstrated that DOXO impinges on cardiomyocyte (CM) signaling pathways that are linked to arrhythmogenesis [18,19,20]. While these and other reports attribute a primary role of CM injury in DOXO-induced arrhythmias, it has to be kept in mind that alterations in the function and topology of cardiac sympathetic neurons (cSNs) may play a relevant role in arrhythmic vulnerability [21,22,23,24,25]. Interestingly, in vitro studies evidenced a direct effect of DOXO on several neuronal populations, including cSNs, through mechanisms that are similar to those that were described for CMs, such as autophagy impairment [26,27], increased oxidative stress [28,29,30,31,32], and DNA damage [33], culminating in neurotoxicity and cell death [30,31,34,35,36].

Although such studies undoubtedly demonstrate that DOXO has direct toxic effects on neurons, the in vitro approach disregards that survival and function of peripheral neurons in vivo depends on intercellular signaling and cell–cell interactions taking place in the innervated microenvironment [37,38]. On this trail, in the effort to understand the interactions between innervating sympathetic neurons (SNs) and cardiac cellular components, we have recently studied in detail the functional and structural interactions between SNs and CMs [39,40,41]. This has led us to demonstrate emerging aspects of neuro-cardiac physiology, reflecting on neuro-cardiac regulation and on the effect of CMs on neuronal health [39,40,41], which laid the hypothesis of the current study. In detail, SNs highly innervate mammalian hearts, including human, with a precise species-specific topology, which, if altered, may cause uneven modification of electrophysiology in discrete heart regions, favoring arrhythmias [25,41]. SN cell bodies mainly organize in cervical ganglia, while their processes invade the myocardial interstitium, with the typical ‘pearl-necklace’ morphology, characterized by regularly distributed varicosities, i.e., the neurotransmitter-releasing sites, which establish, with CMs, a synaptic contact, recently named the neuro-cardiac junction (NCJ) [39,40,41]. At the contact site, a tight intercellular interaction occurs between the neuron and the targeted CM, which underlays both anterograde (SN-to-CM) and retrograde (CM–SN) communication. Through the former, SNs stimulate the heart via noradrenaline which, by activating β-adrenoceptors (β-ARs), enhances heart rate and contractility during stresses [25,39,41], and modulates CM size and electrophysiology, by regulating proteostasis [41,42,43]. In parallel, reverse cardio-neuronal signaling, enacted through nerve growth factor (NGF), provides the continuing trophic input that is required to maintain the functioning and correctly patterned innervation in the developed heart [40]. Indeed, NGF, released from CMs, binds to specific receptors that are expressed on SNs (i.e., TrkA and p75), and the neurotrophin/receptor complexes are retrogradely transported to the neuronal soma, where NGF activates differentiation, survival, and functional signaling pathways [40]. Thus, the refined intercellular interaction, at the NCJ, ties neurons and CMs in a double-stranded bond, which implies that dysfunction of one of such post-mitotic cells negatively impacts on the other. 

On these bases, we tested here the hypothesis whereby the effect of DOXO on CMs may reflect on SN health and, as such, that DOXO negatively impinges on multiple cardiac cell types, including SNs. Confocal immunofluorescence and morphometric analyses in murine and human heart sections have thus been used to assess the state of cardiac sympathetic innervation upon DOXO treatment. Our results show that cardiac innervation is severely compromised by DOXO, which causes a significant decrease in the neuronal density and alterations in innervation topology. In line with our surmised working model, these effects were attributed to the impairment of cardiac neurotrophic signaling by NGF. 

## 2. Results

### 2.1. Doxorubicin Alters Cardiac Sympathetic Neuron Function

To assess whether DOXO treatment affects the function of cSNs, we treated normal adult C57BL/6J male mice with DOXO, following the protocol that was described in the Method section and in Figure 1a. In line with published data, this DOXO regimen caused a progressive decline in body weight during drug administration (Figure 1b), which returned to normal weight within a week from DOXO interruption. In addition, cardiac function, assessed by echocardiography (ECHO) showed declined cardiac contractility six weeks after the end of DOXO treatment (ejection fraction, vehicle: 64.79 ± 5.32 vs. DOXO: 51.43 ± 6.61; fractional shortening, vehicle: 34.40 ± 3.74 vs. DOXO: 25.84 ± 4.16, in %) (Figure 1c,d), which was thus the time point that was chosen for all subsequent histologic and functional heart analyses. 

At this time point, in which the cardiotoxic effects of DOXO were previously reported to manifest [44,45,46], harvested hearts were grossly normal, with slightly expanded interstitial spaces, and moderately infiltrated foci and collagen deposition (Figure 2a,b). Morphometrically, the hearts were moderately atrophic, with a mean decrease in the CM cross-sectional area of 25.40 ± 37.90% (Figure 2c,d).

Electrocardiographic (ECG) recording, performed at different time points before sacrifice, did not evidence differences in the heart rate and QRS duration in anesthetized DOXO- vs. vehicle-treated mice (Figure 3a,b). To determine the functional effects of DOXO on cSNs, the resting SN activity was estimated at the end of experimental protocol by acute administration of atropine, which antagonizes the effects of the parasympathetic branch of the autonomic nervous system on pacemaker cells [42]. While atropine caused the expected effect on heart rate, which increased by 22.43 ± 9.27% in the vehicle-treated mice, DOXO almost completely ablated the atropine-induced heart rate increase (5.68 ± 4.80%), suggesting dysfunction in the neurogenic control of heart rhythm, consistent with heart sympathetic denervation (Figure 3c,d). 

### 2.2. Doxorubicin Compromises the State of Sympathetic Innervation in Murine Hearts

The evidence of decreased chronotropic effect of SNs in DOXO-treated mice prompted us to assess whether such functional failure was underscored by sympathetic neuropathology. To this purpose, non-consecutive heart sections from the mid-portion of the ventricles were analyzed by confocal immunofluorescence, and SN processes were identified using an anti-tyrosine hydroxylase antibody. Normal mouse hearts are highly innervated by SNs, which distribute in the myocardial interstitium showing regularly displaced varicosities along the neuronal process [40]. Qualitative analysis of the heart from the DOXO-treated mice suggested that the density of cardiac sympathetic innervation was profoundly decreased in both the right and left ventricles (Figure 4a). This evidence was confirmed by morphometric analysis of the innervation density in the left ventricle (LV), demonstrating an 82.22 ± 13.93% decrease in the fraction of myocardial area that was occupied by tyrosine hydroxylase-positive fibers (Figure 4b,c). Such denervation was of a similar degree in both the sub-epicardial and sub-endocardial regions (sub-epicardium, vehicle: 3.11 ± 0.76 vs. DOXO: 0.58 ± 0.45; sub-endocardium, vehicle: 2.15 ± 0.45 vs. DOXO: 0.35 ± 0.26, in %) (Figure 4c), which we have recently demonstrated to be innervated at different density, reflecting the peculiar electrophysiologic and trophic effects of neuronal activity in the two regions [41]. 

In both the sub-endocardial and sub-epicardial regions, SN processes of DOXO-treated hearts appeared thinner and fragmented, with a significant reduction in the area of varicosities (Figure 4d,e) and immunoreactivity to anti-tyrosine hydroxylase (fluorescence intensity, vehicle: 45.31 ± 4.71 vs. DOXO: 13.89 ± 8.18, a.u.), all features that are suggestive of sympathetic neurodegeneration.

Based on these results, we harvested and analyzed superior cervical and stellate ganglia, which contain the cell soma of most cSNs, from both the control and DOXO-treated mice. Confocal immunofluorescence analysis demonstrated that the DOXO-treated ganglia had a significant reduction in the density of neuronal soma (Figure 5a,b) and a decrease in the average size of the remainder, which were atrophic (Figure 5c).

### 2.3. Doxorubicin Treatment Reduces Heart-Derived NGF Input to Neurons

The evidence that, in DOXO-treated mice, cSN varicosities were smaller in size and SN cell bodies were atrophic are all features that were previously linked to reduced neurotrophic input, which we have shown to rely on NGF that is released from innervated CMs [40]. This hypothesis is in line with the previous demonstration that DOXO treatment severely compromises NGF production in rat hearts [47]. In line with this study, our results showed that the NGF protein content in hearts trended to decrease, since three days after initiating DOXO treatment, and become significantly lower at three days after completion of the drug regimen up to six weeks (Figure 6a,b). Consistently, immunofluorescence with an anti-NGF antibody in SN of cervical ganglia sections, revealed decreased immunoreactivity in the DOXO-treated samples, supporting defective retrograde CM–SN signaling via NGF (Figure 6c). 

### 2.4. Doxorubicin Compromises Sympathetic Neurons in Human Hearts

The results that we acquired in the experimental models suggest that the broad spectrum of manifestations of DOXO cardiotoxicity may include the degeneration of cSNs, an aspect that is potentially linked to heart dysfunction (e.g., arrhythmias), that has never addressed in detail thus far. To ascertain whether neuropathologic features, similar to those that were observed in mouse hearts could be detected in cSN processes of patients that had undergone DOXO treatment, we analyzed sections of explanted hearts from three patients who developed HF upon chemotherapy cardiotoxicity (Figure 7a). In these samples, confocal immunofluorescence demonstrated a global decrease of myocardial innervation density and fragmentation of neuronal processes (Figure 7b,c). Our results surmise that morphologic and functional alterations in cSNs, resulting in denervation and loss of neuronal inputs to cardiac cells, are additional and previously neglected effects of DOXO treatment. 

## 3. Discussion

In this study, by combining in vivo assessment of cardiac function and analyses of heart samples from mice that had undergone DOXO administration, with that of hearts that were explanted from cancer-bearing patients that were suffering post-chemotherapy HF, we show that cSNs are affected by DOXO treatment. Our results suggest that cSN degeneration is accompanied with reduced cardiac NGF production, likely compromising ‘CM-neuron’ neurotrophic input, which we previously showed to be essential to maintain correct cardiac sympathetic innervation [40]. These results identify a previously neglected mechanism of DOXO-induced neurodegeneration, potentially synergizing with other direct effects of the drug on neurons, including autophagy impairment, DNA damage, and increased oxidative stress [26,28,29,31,32,33]. In addition, we uncover the neuropathologic effect of DOXO on SNs regulating heart function, which may be involved in HF and arrhythmias, both distinctive features of DOXO cardiotoxicity [4,13,48,49,50,51].

Although DOXO is the most common chemotherapeutic agent in several malignancies, its use is associated with important cardiac complications, enclosed in the broad definition of “DOXO cardiotoxicity” [52,53,54,55]. Based on the time of onset, cardiotoxicity may be acute, manifesting within two weeks from the end of treatment; chronic, developing within one year; or late-onset, developing even several years after chemotherapy completion. In addition, subclinical DOXO cardiotoxicity has increasingly been described, and prompts the identification of criteria and biomarkers for early diagnosis and treatment [51,56,57,58,59,60]. Cardiac complications of DOXO represent a high socio-economic burden, as they may lead to HF, the first cause of death/transplant in cancer survivors, myocardial infarction, and arrhythmic episodes (i.e., tachycardia, atrial and ventricular arrhythmias) [12,50,61,62,63,64,65]. In addition, such complications, when occurring acutely during treatment, may require chemotherapy interruption with inauspicious outcomes for patients, while, in the long-term, they may severely compromise patient life quality and expectancy, as in the case of heart transplant (for HF) or implantable cardioverter-defibrillator (ICD) implantation (for arrhythmias) [4,61,62,63,65,66,67]. Thus, preventing/reducing the risks of cardiotoxicity and its progression, without reducing the efficacy of anti-cancer treatment, is a primary goal of cardio-oncology.

Up to now, the term ‘cardiotoxicity’ has been used as synonym of ‘cardiomyocyte (CM) toxicity’ and, as such, cardio-oncology has mainly focused on CMs, which are undoubtedly affected by DOXO [53,54,55,68,69,70,71,72,73]. Abundant research has attributed the adverse cardiac effects of DOXO to a plethora of mechanisms affecting cell electrophysiology, signaling, mechanics, and cell survival, which subsequently lead to myocardial remodeling, contractile dysfunction, or arrhythmias [4,18,54,55,74,75]. However, some manifestations of cardiotoxicity (e.g., arrhythmic episodes, alterations in ECG parameters) may involve the impairment of cardiac extrinsic regulatory systems, including e.g., the altered function or distribution (i.e., myocardial topology) of cSNs [76,77,78,79,80]. 

The heart is highly innervated by SNs, whose activity has conventionally been associated with heart adaptation to acute stresses, during the ‘fight-or-flight’ reaction [25]. Recently, we have uncovered additional long-term effects of cSNs which, by regulating proteolytic machineries via β-ARs, determine CM size and electrophysiology [25,41,42]. The microscopic interactions between neurons and target cardiac cells have recently been characterized in higher detail, and the concept of direct neuro-cardiac communication, as underscored by specific intercellular synaptic contacts has been appraised [39,40,81]. Such contact sites have been identified as a hub of adrenergic signaling ‘from-neuron–CMs’, shaping and controlling target cell trophism and function, on the one hand [39,41], and, on the other, of neurotrophic signaling ‘from-CMs–neurons’, determining and maintaining the topology of the myocardial innervation network [40]. Such local and spatially determined neuro-cardiac interactions allow the activation of cSNs to result in adequate cardiac responses, while avoiding arrhythmias. Bidirectional SN-CM coupling implies that failure of CMs to support the innervating neurons with NGF would result in neurodegeneration, local cardiac denervation, and aberrant innervation topology, the latter potentially ensuing in arrhythmias [76,77,78,79,80]. Neurodegeneration may, in turn, compromise cardiac adrenergic signaling, which reflects on CM structure and function, jeopardizing NGF production, and in a vicious cycle, worsen sympathetic denervation. 

On these bases, we here approached DOXO cardiotoxicity as a condition affecting the entire myocardial cell network and focused on the effects of DOXO on cardiac innervation, as explored in the hearts of mice that were exposed to DOXO in a chemotherapy regimen that was similar to that which is used in humans [14,82]. Our data demonstrate that DOXO causes severe cardiac sympathetic denervation, accompanied with alterations in neuronal distribution pattern in both ventricles, as shown by immunofluorescence, and function, as indicated by the reduced heart rate increase, upon atropine administration in DOXO mice. Interestingly, the reduced SN inputs to target CMs is predicted to impinge on signaling pathways and cell homeostatic systems (e.g., autophagy, oxidative stress, Ca^2+^ dynamics) which have been described to be independently affected by DOXO [14,42,53,82,83]. It can thus be inferred that the toxicity of DOXO on CMs may be amplified by its adverse effects on cSNs. As an example, the atrophic remodeling of DOXO-treated CMs may be the consequence of activated intracellular proteolysis, by the ubiquitin proteasome system, combined with the indirect effect of SN degeneration, further promoting CM induction of atrogenes through decreased β2-AR stimulation [42]. In a similar manner, SN degeneration may result from a cell-directed effect of DOXO on neurons, the secondary effect of toxicity on CMs, or a combination of both mechanisms. In line with recently published data [47], we observed that hearts from DOXO-treated mice have reduced NGF content, already in the initial phases of treatment, which is expected to compromise the ability of CMs to feed innervating neurons. Such mechanisms, previously not taken into account, may overlap the neurotoxic effects of DOXO, including the induction of oxidative stress and autophagy impairment, given that NGF, in addition to activating pro-survival and trophic signaling [84,85,86,87], has antioxidant effects and is an autophagy activator in both CMs and SNs [88,89,90,91]. 

Our data identifies new cellular targets and systems that are affected by DOXO. While this finding seemingly increases the complexity of the pathogenetic framework underlying DOXO cardiotoxicity, it also directs therapeutic strategies to comprehensively treat CMs, cSNs, and the neuro-cardiac axis. The inference of our results would suggest approaches that are based on NGF targeting, i.e., to increase cardiac NGF availability. Such an approach, however, is complicated by the multitude of effects that are exerted by the neurotrophin on different cell types, including cancer cells, whose treatment with a growth factor is in obvious contrast with the general goal of chemotherapy [92,93,94,95,96]. Interestingly, what emerges from our results and consolidated published data, is that most mechanisms leading to cardiotoxicity, including the alteration of neuro-cardiac regulation, converge on autophagy impairment and the activation of oxidative stress. This is in line with the lead concept supporting current research efforts, which target autophagy and oxidative stress in DOXO cardio-protection, bringing into the scenario the idea that therapeutic approaches need to be directed to specific regulatory mechanisms in different cell types, not to the sole CMs.

The finding that cSNs are targeted by DOXO is in line with existing clinical evidence and has implications at different levels, spanning from basic knowledge on disease mechanisms, research pursuing the identification of therapeutic strategies in DOXO-treated patients, to the definition of improved diagnostic tools for the management of cancer patients. 

From *the preclinical standpoint*, our evidence of cardiac denervation, obtained in experimental rodents and human heart samples, confirmed previous case reports in which nuclear imaging revealed cSN atrophy and denervation in patients with DOXO-cardiomyopathy. Interestingly, both in humans and in mice, it was surmised that SN degeneration is a precocious event preceding cardiac adverse symptoms [97,98], suggesting that hearts are exposed to progressively variable, and possibly heterogeneous or erratic SN activity, until denervation. Although with the obvious limitation of a proof-of-concept study and those of individual case reports, this data broadens the spectrum of cells and signaling pathways that are affected by DOXO treatment, increasing the knowledge on the basic mechanisms of DOXO cardiotoxicity. In addition, these results laid the ground for further research that is aimed at assessing the state of sympathetic innervation of other organs and districts, e.g., gut, skeletal muscles, which may explain extracardiac manifestations of DOXO toxicity (e.g., modification of gut microbiota, tissue inflammation) [99,100,101]. 

Our results may also guide towards the identification of *mechanism-driven therapie*s in DOXO-treated patients. Indeed, given that cardiac sympathetic dysfunction has previously been linked to arrhythmias and HF, both of which are frequently experienced by patients, understanding the time- and dose-dependency of DOXO effects on cSNs is key for refining therapy (including β-AR blockers and neuromodulation) to prevent such adverse consequences. Prospectively, future implications may include use of neuro-protective approaches to alleviate cardiac and the systemic side effects of DOXO. 

Finally, this study may stimulate *research on additional criteria and biomarkers* to be considered for the early detection and optimal management of DOXO cardiotoxicity, including functional (e.g., autonomic function assessment), molecular (e.g., biochemical: noradrenaline, neuropeptide-Y plasma levels), and histopathologic (e.g., sympathetic innervation of skin biopsies) autonomic tests. 

## 4. Materials and Methods

### 4.1. Human Samples

Here, we analyzed sections from human subjects who died for non-cardiac causes (n = 3) and subjects who underwent a chemotherapy regimen including DOXO and developed cardiomyopathy (n = 3). Ventricular samples were acquired during routine post-mortem investigations or during post-transplant evaluation. Then, the samples were archived in the anatomical collection of the Institute of Pathological Anatomy of the University of Padova. The samples were anonymous to the investigators and used in accordance with the “Recommendation CM/Rec (2016) of the Committee of Ministers of member States on research on biological materials of human origin”, released by the Council of Europe, as received by the Italian National Council of Bioethics. The samples were analyzed using protocols previously described [102].

### 4.2. Animal Models

In this study, we used adult C57BL/6J male mice (Charles River, Milan, Italy). The animals were maintained in authorized animal facilities (authorization number 175/2002A), at controlled temperature, with a 12-on/12-off light cycle and had access to water and food ad libitum. All the experimental procedures that were performed on rodents were approved by the local ethical committee and the Ministry of Health (Authorization numbers 408/2018PR, 129/2018PR and 738/2016PR), in compliance with Italian Animal Welfare Law (D.Leg 4/3/2014 and subsequent modifications). All procedures were performed by trained personnel with documented formal training and previous experience in experimental animal handling and care. All protocols were refined prior to starting the study, and the number of animals was calculated to use the least number of animals that was sufficient to achieve statistical significance according to sample power calculation. 

### 4.3. In Vivo DOXO Treatment

Two months (mo.) old C57BL/6J male mice were injected with either DOXO (3mg/kg, i.p., Tocris, Bristol, UK) or vehicle solution (sterile water) on alternate days for 14 days, as described in (Figure 1a). This regimen was associated to an index of mortality that was lower than 5%. At the end of treatment, mice underwent electrocardiographic (ECG) and echocardiographic (ECHO) analyses at the time points indicated in Figure 1a. At six weeks after the completion of the DOXO treatment, the mice were sacrificed by cervical dislocation. The hearts and superior cervical/stellate ganglia were harvested, washed in 1X PBS, and processed for molecular and IF/histological analyses as previously reported [40,103]. Samples to be analyzed by IF were fixed in 1% paraformaldehyde for 30 min, dehydrated in sucrose gradient, and frozen in liquid nitrogen.

### 4.4. Echocardiographic Analysis

ECHO was performed as previously described [103], in mice that were anesthetized with isoflurane (2.5% *v*:*v* in O_2_) during constant monitoring of temperature, heart, and respiration rates and ECG parameters. The animals were imaged using a Vevo 2100 system (Fujifilm VisualSonics, Toronto, Canada), that was equipped with a 30-MHz transducer. Briefly, two-dimensional cine loops with frame rates of 200 frames per second of a long-axis view and a short-axis view at proximal, mid, and apical level of the LV were recorded. The ejection fraction (EF) was determined by the following formula, based on the Simple method (Simp): %EF = 100 × systolic LV volume/diastolic LV volume.

### 4.5. Electrocardiographic Analysis

The mice were anesthetized with isofluorane (2.5% *v*:*v* in O_2_) and ECG was recorded by Powerlab 8/30, Bioamp (from AD Instruments, Dunedin, New Zeland) both at baseline and upon atropine administration (2mg/kg, i.p.), as described in [104]. Heart Rate and standard ECG parameters were calculated using the software LabChart 8 (AD Instruments, Dunedin, New Zealand).

### 4.6. Confocal Immunofluorescence in Murine Samples

Ten-μm heart or stellate/cervical ganglia cryosections were obtained with a cryostat (CM1860; Leica, Wetzlar, Germany) and processed for IF analysis as previously reported [102]. Briefly, heart or ganglia cryosections were incubated O/N at 4 °C with primary antibodies that were diluted in PBS, supplemented with 1% Bovine Serum Albumin and 0.5% Triton-X100 (all from Sigma-Aldrich, St. Louis, MO, USA). Cryosections were then incubated with secondary antibodies for 30 min at 37°C. The primary and secondary antibodies that were used in this study are listed in Table 1. The images were acquired with a confocal microscope that was equipped with a 63x objective (1.4 NA) (Zeiss LSM900, Carl Zeiss, Oberkochen, Germany), and used for the morphometric analyses that were described in 4.8.

### 4.7. Confocal Immunofluorescence in Human Samples

The samples were analyzed using the protocol that was previously described in [102]. Briefly, 3 μm thick heart sections were unmasked using microwave irradiation. The sections were permeabilized with 10% Triton X-100 (Sigma-Aldrich, St. Louis, MO, USA) for 2 h at 37 °C, and incubated O/N with the appropriate primary antibody (Table 1).

### 4.8. Morphometric Evaluation of Cardiac Innervation Density

Neuronal density and CM cross-sectional areas were calculated in six non-consecutive cryosections from the mid-portion of the ventricles of vehicle- and DOXO-treated mice. For each section, six images from both the subepicardial and subendocardial regions, identified as described in [41], were acquired and analyzed using the software ImageJ (version 1.53q, National Institutes of Health, Bethesda, MD, USA). In detail, a z-projection of the stack was obtained, and neuronal density was calculated as the percentage of tyrosine hydroxylase-positive area over the area of epi- or endo- regions that were analyzed. CM cross-sectional areas were calculated as previously described [41].

### 4.9. Morphometric Evaluation of Cardiac Fibrosis

The percentage of fibrosis in human heart samples was calculated automatically on a virtual-colour-based system, and reported as the mean value of 10 different randomly chosen fields [105].

### 4.10. Protein Extraction and Western Blotting Analysis

The analysis was performed as previously described [103]. In detail, the hearts were lysed on ice for 15 min in 120 mM NaCl, 50 mM Tris-HCl (pH 8.0), 1% Triton X-100, protease inhibitor Complete (Roche Applied Science, Penzberg, Germany), and phosphatase inhibitors (50 mM sodium fluoride, 1 mM sodium orthovanadate, and 10 mM sodium pyrophosphate). The lysates were cleared by centrifugation at 13,000 rpm for 15 min at 4 °C. The protein concentration was determined by the Bradford method. The proteins from hearts or cellular lysates were separated by SDS-polyacrylamide gel electrophoresis (SDS-PAGE) and transferred to methanol-activated polyvinylidene difluoride (PVDF) membranes (Millipore Corporation, Billerica, MA, USA). The membranes were incubated for 1 h with 5% bovine serum albumin (BSA)-TBST [tris-buffered saline (TBS)-0.3% Tween 20] at room temperature and overnight incubated with primary antibodies at 4 °C. Appropriate host species horseradish peroxidase-conjugated secondary antibodies were added and signals were detected with enhanced chemiluminescence (Millipore Corporation, Billerica, MA, USA). The antibodies that were used for this analysis are listed in Table 2.

### 4.11. Statistics

Statistical analysis was performed using GraphPad Prism 8. The normality of data distribution was assessed with a Shapiro–Wilk test. An unpaired *t*-test (for two groups) or one-way ANOVA (for three or more groups) were used for normally distributed data. An unpaired *t*-test with Welch’s correction was used to compare two groups with normally distributed data and unequal variance. A Mann–Whitney test was used to compare two groups with non-normally distributed data. A *p*-value < 0.05 was considered statistically significant.

## Figures and Tables

**Figure 1 ijms-23-11098-f001:**
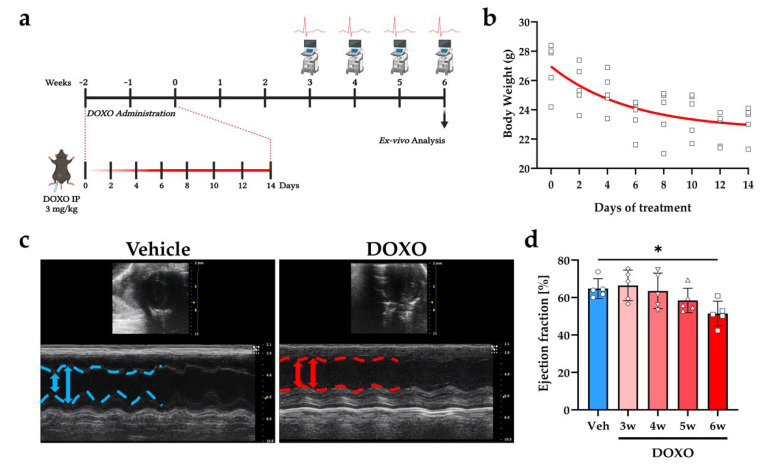
DOXO treatment leads to cardiac contractile dysfunction. (**a**) DOXO administration and ex vivo analyses protocol: study design. (**b**) Body weight evaluation during mouse treatment with DOXO. The nonlinear fitting line is shown in red. (**c**) Echocardiographic short-axis (top) and B-mode (bottom) views of hearts from vehicle- vs. DOXO-treated mice. These analyses were performed six weeks after completion of drug administration regimen. Blue/red lines and arrows represent left ventricle (LV) systolic and diastolic diameters in vehicle- and DOXO-treated mice, respectively. (**d**) Echocardiographic assessment of LV ejection fraction in vehicle- and DOXO-treated mice, at different time points after completion of DOXO administration. Bars represent the standard deviation (s.d.). Differences among the groups were determined using one way ANOVA with Dunnett’s test for multiple comparisons. (*n* = 5 mice for each group; *, *p* < 0.05). DOXO, doxorubicin; Veh, vehicle.

**Figure 2 ijms-23-11098-f002:**
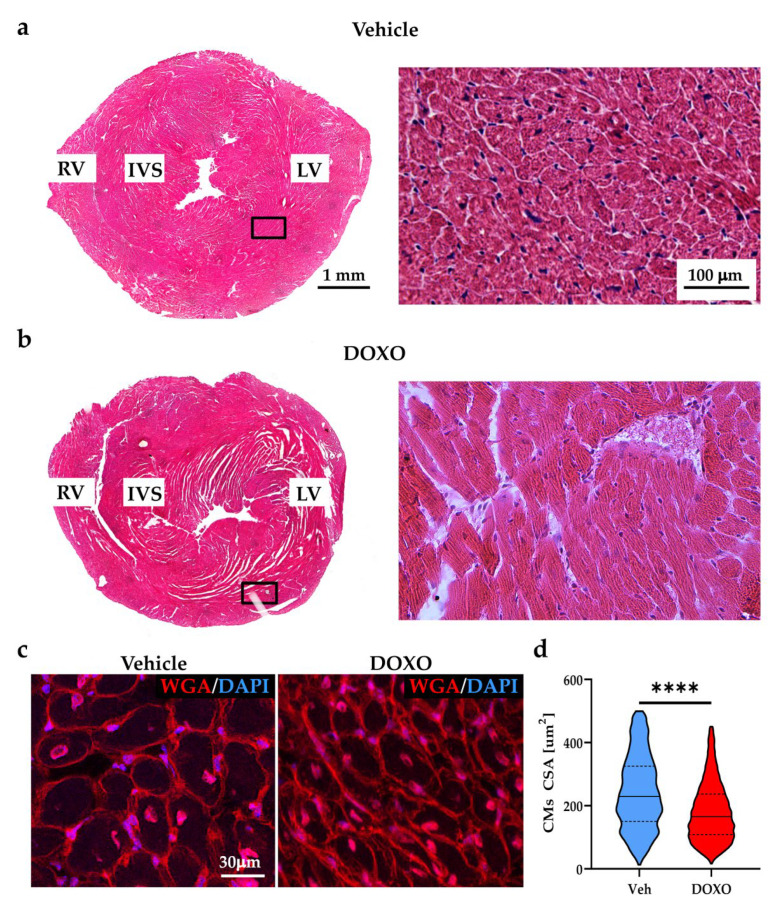
DOXO treatment leads to cardiac atrophic remodeling. (**a**,**b**) Hematoxylin-eosin staining in heart sections from the vehicle- (**a**) and DOXO- (**b**) treated mice, six weeks after the completion of DOXO administration. Right panels in (**a**,**b**) are high magnification of the black boxed areas in the left panels. (**c**) Confocal immunofluorescence on heart sections from the mid-portion of the ventricles of vehicle- vs. DOXO-treated mice, stained with Alexa Fluor™-555-conjugated wheat germ agglutinin (WGA). The nuclei were counterstained with 4′,6-Diamidino-2-Phenylindole (DAPI). Images were used to calculate the cardiomyocyte (CM) cross-sectional areas (CSA) (**d**). Differences among the groups were determined using a Mann–Whitney test. (n > 2500 CMs from four different hearts for each group; ****, *p* < 0.0001). The solid line on violin plot represents the median, dashed lines are 25/75 percentiles. RV, right ventricle; IVS, interventricular septum; LV, left ventricle; DOXO, doxorubicin; Veh, vehicle.

**Figure 3 ijms-23-11098-f003:**
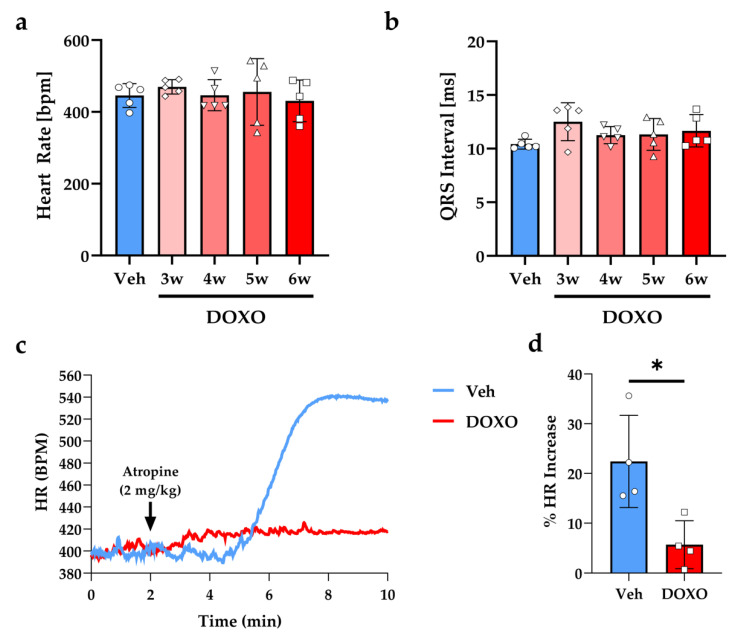
DOXO treatment affects cardiac sympathetic neuron function. (**a**,**b**) Electrocardiographic evaluation of heart rate (HR, (**a**)) and QRS duration (**b**) in the DOXO- vs. vehicle-treated mice, at different time points after the completion of DOXO administration. (**c**) Representative trace of HR increase in a vehicle- vs. a DOXO-treated mouse upon Atropine injection. (**d**) Evaluation of the fractional HR increase 5 min after atropine administration. Bars represent the s.d. Differences among the groups were determined using an unpaired *t*-test. (n = 4 mice for each group; *, *p* < 0.05). DOXO, doxorubicin; Veh, vehicle; BPM, beats per minute.

**Figure 4 ijms-23-11098-f004:**
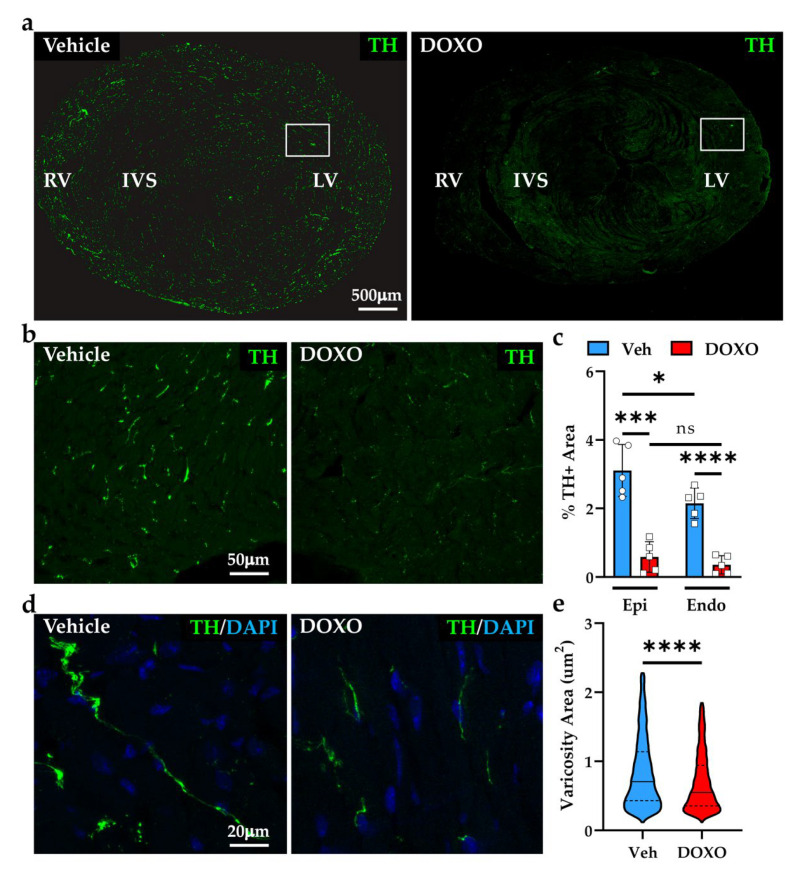
DOXO treatment causes cardiac sympathetic denervation in mice. (**a**) Confocal immunofluorescence on heart sections from the ventricular mid-portion of vehicle- vs. DOXO-treated mice. Sections were stained with an antibody to tyrosine hydroxylase (TH). (**b**) Images are high magnification of the white boxed areas in (**a**). (**c**) Quantification of the percentage of TH-positive area in the sub-epicardial (Epi) and sub-endocardial (Endo) LV myocardium of the control (Veh) and DOXO hearts. Each point represents the average tyrosine hydroxylase positive (TH+) area of 20 independent images from the Epi and Endo regions in five hearts analyzed/group. Bars represent the s.d. Differences among the groups were determined using an unpaired *t*-test (*, *p* < 0.05; ***, *p* < 0.001; ****, *p* < 0.0001; ns, no significant). (**d**) High magnifications of SN processes in the LV of control and DOXO hearts. (**e**) Quantification of sympathetic neuron (SN) varicosity area. Differences among the groups were determined using a Mann–Whitney test. (n > 1000 varicosities from five different hearts in each group; ****, *p* < 0.0001). RV, right ventricle; IVS, interventricular septum; LV, left ventricle. DOXO, doxorubicin; Veh, vehicle; DAPI, 4′,6-Diamidino-2-Phenylindole.

**Figure 5 ijms-23-11098-f005:**
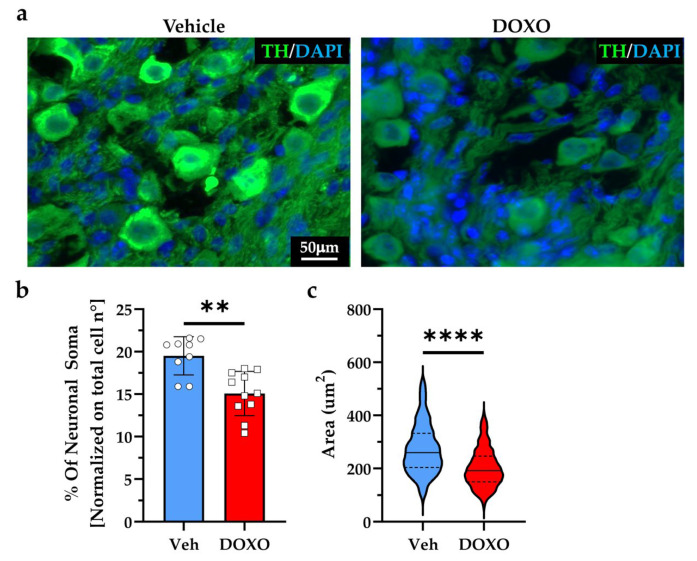
DOXO causes cardiac sympathetic neuron degeneration. (**a**) Immunofluorescence on stellate ganglia sections from the vehicle- vs. DOXO-treated mice. Slices were stained with an antibody to tyrosine hydroxylase (TH). Nuclei were counterstained with 4′,6-Diamidino-2-Phenylindole (DAPI). (**b**,**c**) Quantification of the fraction of neuronal soma/total cell number (**b**) and the mean area of the sympathetic neuron (SN) cell bodies (**c**) in the stellate ganglia of the DOXO- vs. vehicle-treated mice. Bars represent the s.d. Differences among the groups were determined using a Mann–Whitney test. (n > 500 cells from nine independent images/group from three different mice; **, *p* < 0.01; ****, *p* < 0.0001). DOXO, doxorubicin; Veh, vehicle.

**Figure 6 ijms-23-11098-f006:**
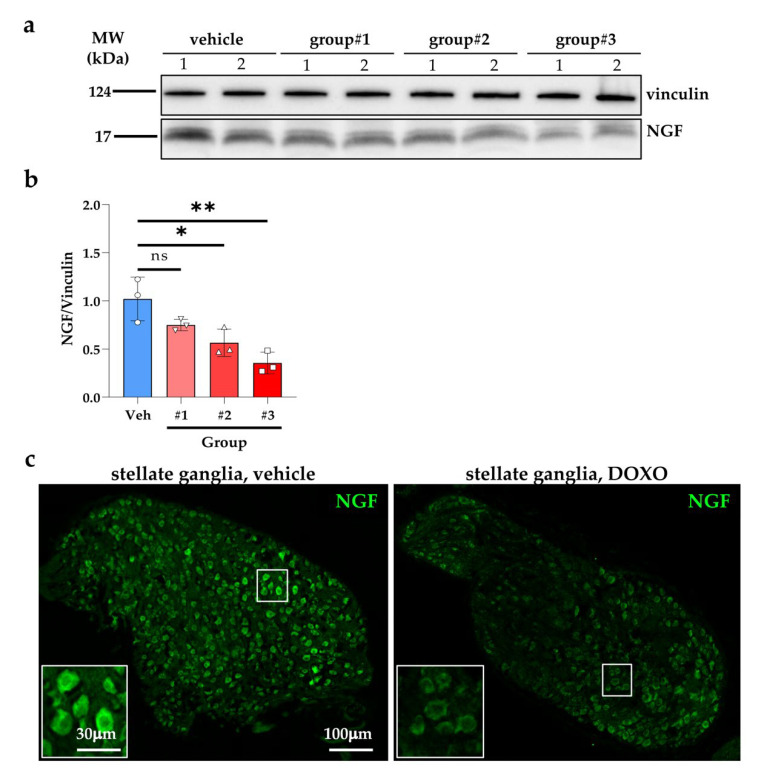
DOXO treatment is accompanied by reduced cardiac NGF content. (**a**) Western blotting on the protein extracts of hearts that were explanted from: control mice (vehicle-treated); mice that were treated with DOXO for three days (group#1); mice that were analyzed three days after the completion of DOXO treatment (group#2); mice that were analyzed 6 weeks after the completion of DOXO administration (group#3). (**b**) The relative densitometry is shown in the right graph. Bars represent the s.d. Differences among the groups were determined using a one-way ANOVA with Dunnett’s test for multiple comparisons. (n = 6 hearts for each group; *, *p* < 0.05; **, *p* < 0.01, ns, statistically non-significant). (**c**) Immunofluorescence on stellate ganglia sections from vehicle- vs. DOXO-treated mice, six weeks after completion of DOXO treatment. Sections were stained with an antibody to Nerve Growth Factor (NGF). DOXO, doxorubicin; Veh, vehicle.

**Figure 7 ijms-23-11098-f007:**
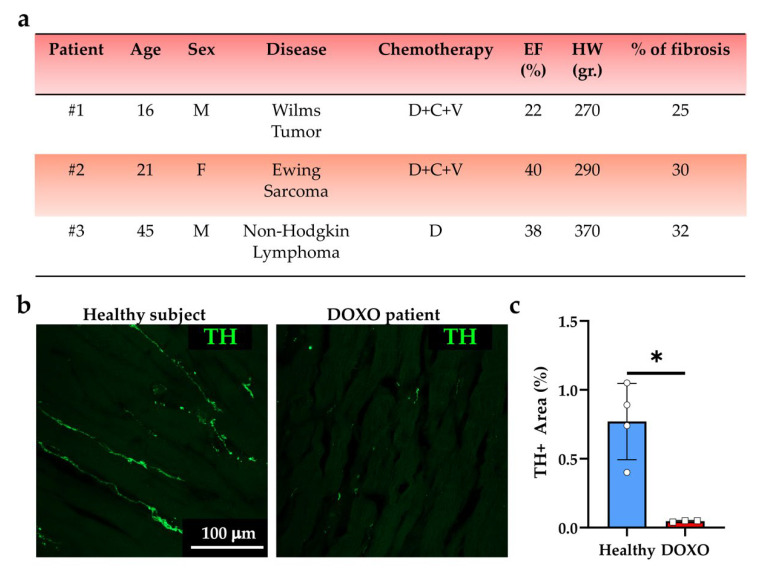
Patients receiving DOXO treatment show cardiac sympathetic denervation. (**a**) Characteristics of patients that were analyzed in the study. D, DOXO; C, cytarabine; V, vincristine. The percentage of fibrosis was calculated as described in the Methods section. Heart weight was assessed at the time of explant. (**b**) Confocal immunofluorescence on human heart sections from the mid-portion of the LV in healthy vs. DOXO-treated hearts. Sections were stained with an antibody to tyrosine hydroxylase (TH). The images were used to quantify the percentage of myocardial area that was occupied by TH-positive (TH+) fibers (**c**). Bars represent s.d. Differences among groups were determined using unpaired *t*-test with Welch’s correction. (Four (Ctrl) and three (DOXO) independently processed heart slices from three different hearts/groups were analyzed; *, *p* < 0.05).

**Table 1 ijms-23-11098-t001:** List of primary and secondary antibodies that were used for immunofluorescence analysis.

Target	Host	Company	Ref. Number	Dilution
Tyrosine Hydroxylase	Rabbit	Millipore	Ab152	1:400
Nerve Growth Factor	Rabbit	Alomone	AN-240	1:100
WGA-Alexa Fluor™-555	None	Invitrogen	W32464	1:400
Anti-Rabbit-488	Goat	Jackson Lab.	111-545-144	1:200

**Table 2 ijms-23-11098-t002:** List of primary antibodies used for biochemical analyses.

Target	Host	Company	Ref. Number	Dilution
Nerve Growth Factor	Rabbit	Alomone	AN-240	1:200
Vinculin	Rabbit	Cell Signaling	4650	1:1000

## Data Availability

Data available upon request.

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
