# Peer review of "Neurotoxic Effect of Doxorubicin Treatment on Cardiac Sympathetic Neurons"

_ijms, 2022, doi:10.3390/ijms231911098_

Round 1

Reviewer 1 Report

This is an experimental report regarding cardiac sympathetic neurons (SNs) and doxorubicin (DOXO), which aimed to examine DOXO cardiotoxicity with a particular focus on SNs, with the hypothesis that they might represent additional cell types affected by DOXO and potential targets to be looked at to alleviate cardiotoxicity. The authors demonstrated the contribution of cSNs to chemotherapy cardiotoxicity, unveiling a previously unrecognized effect of DOXO on cardiac autonomic regulation, which was involved in both cardiac physiology and pathology, including heart failure and arrhythmias. 

This reviewer considers that the authors have well performed this study and has some comments as described below. 

Major comments:

1.     The aim of this study was relatively unclear. The authors should describe their aim more clearly at the end of Introduction section. 

2.     Figure 1. Blood pressure and heart rate data should be added during the observation period.  

3.     How about mortality in these animals? Were there any deaths caused by arrhythmia? 

4.     The aim was not clearly described in the Introduction section, and further, the conclusion was also unclear, which was described at the end of the Discussion section. The authors should describe more clearly. And, there were 2 “Finally” sentences. Which was the final description? 

Author Response

Dear Reviewer,

attached you can find the pdf version of the 'point-by-point' response.

With best reagrds,

Tania Zaglia

Marco Mongillo

Reviewer 2 Report

The authors assessed neurotoxic effect of doxorubicin treatment on cardiac sympathetic neurons. Their data supported the contribution of cSNs to chemotherapy cardiotoxicity, unveiling a previously unrecognized effect of DOXO on cardiac autonomic regulation.

I have the following concerns:

1. The article should be composed as follows: introduction, methods, results, conclusions.

2. Small sample size limits the study.

3. What are the practical implications of the study?

4. What was the echocardiographic protocol, type of ultasound machine, methods of LVEF measurement?

Author Response

(The authors gave the same response as above.)

Reviewer 3 Report

The manuscript by Moro N. et al. addresses the old but still unresolved topic of Doxorubicin (DOXO)-induced cardiotoxicity. Beside the cardiomyocyte (CM) death, the authors proposed cardiac sympathetic neurons (cSNs) as novel target cell type involved in DOXO-induced arrhythmias. In the heart CMs and cSNs are functional and structural related to each other, in particular Nerve Growth Factor (NGF) secreted by cardiomyocytes is necessary to maintain SNs in health and function. Using an in vivo approach, the authors investigated whether cardiac innervation by SNs and cardiac NGF level are impaired by DOXO. The data showed alterations in terms of innervation density, topology, and function of SNs in murine hearts, analysed several weeks after the DOXO therapy had stopped. Interestingly, these changes in myocardial innervation resembled those observed in cardiac biopsy from DOXO treated patients. The gradual and long-lasting decreased of Nerve Growth Factor (NGF) was considered an important factor impacting the innervation in heart of DOX treated mice.

Overall, the experimental research is appropriated designed but additional analysis are needed to strengthen some conclusions:

1)      In immunofluorescence images the intensity of TH staining was always lower in DOXO treated samples. Can the authors explain this result? Can SNs be identified with an alternative marker?

2)      The decrease of NGF in Western Blot of Fig.6 a is not convincing. Please include more samples in the analysis. The data can be also supported by the quantification of immunofluorescence images.  

Moreover, the evaluation of NGF gene expression should be included.

3)      NGF binds to the receptors TrkA and p75NTR triggering the activation of specific downstream signalling pathways. Is the activation or level of these receptors impaired by DOXO-induced reduced NGF? In addition to activating pro-survival and trophic signalling, NGF has antioxidative effects and it is an autophagy activator in both CMs and SNs. Can the authors investigate on these aspects?

Minor points

1)      The conclusion referred to Fig. 5 “DOXO-treated ganglia had a significant reduction in the density of neuronal soma and a decrease in the average size of the remainder, which were atrophic” line 189-190 page 6) does not fit with the image and the text to figure. The quantification is referred to in (b) to smaller soma and (c) to neuronal density. Please rephrase for more clarity.

2)      In text to figure Fig. 7a and in Material and methods is not indicated how EF, HW, % fibrosis were calculated. Please add the details.

3)      Introduction: more information about NGF and its role on SN are necessary for the readers to emphasize the cardiac-neuron interdependence.

4)      Materials and methods: even if the procedures are summarized as “previously described”, a brief description is needed.

Author Response

(The authors gave the same response as above.)

Round 2

Reviewer 1 Report

This reviewer has no further comment. 

Reviewer 2 Report

I have no further comments. Thank you. 

Reviewer 3 Report

The authors properly replayed and modified the manuscript as suggested.